# Minimax Dynamics of Optimally Balanced Spiking Networks of Excitatory and Inhibitory Neurons

**Qianyi Li**
Biophysics Graduate Program
Harvard University
Cambridge, MA 02138
qianyi_li@g.harvard.edu

**Cengiz Pehlevan**
John A. Paulson School of Engineering and Applied Sciences
Harvard University
Cambridge, MA 02138
cpehlevan@seas.harvard.edu

## Abstract

Excitation-inhibition balance is ubiquitously observed in the cortex. Recent studies suggest an intriguing link between balance on fast timescales, tight balance, and efficient information coding with spikes. We further this connection by taking a principled approach to optimal balanced networks of excitatory (E) and inhibitory (I) neurons. By deriving E-I spiking neural networks from greedy spike-based optimizations of constrained minimax objectives, we show that tight balance arises from correcting for deviations from the minimax optima. We predict specific neuron firing rates in the networks by solving minimax problems, going beyond statistical theories of balanced networks. We design minimax objectives for reconstruction of an input signal, associative memory, and storage of manifold attractors, and derive from them E-I networks that perform the computation. Overall, we present a novel normative modeling approach for spiking E-I networks, going beyond the widely-used energy-minimizing networks that violate Dale's law. Our networks can be used to model cortical circuits and computations.

## 1 Introduction

While spiking neural networks empower our brains, a thorough understanding of how spikes accurately represent and transmit information is lacking. Any explanation should address a striking and widely observed property of cortical spiking networks that excitatory (E) and inhibitory (I) currents for individual neurons are balanced (detailed balance), and spikes are only generated during the brief occasions when inhibition fails to track spontaneous fluctuations in excitation (tight balance) [1–8]. In this work, we show that tight and detailed balance could arise from correcting for deviations from a minimax optimum that governs the dynamics of E-I balanced networks. Having access to an objective allows us to design spiking E-I networks that perform various biologically relevant functions.

While the theory of balanced networks goes further back [9–11], the idea that tight and detailed balance is related to the efficiency of spike coding was proposed in recent influential theoretical work [6, 12, 13]. These results are typically limited to spiking dynamics performing a greedy-minimization of a reconstruction error loss function without considering more diverse computations carried out in the cortex. Moreover, minimizing an energy function leads to symmetric connection weights [14] that violate, among other biological constraints, Dale's law: a neuron's influence on its post-synaptic neurons are either all excitatory or all inhibitory. This violation has been addressed by introducing separate reconstruction error cost functions for E and I neurons [12,15]. While E neurons reconstruct an input signal, I neurons reconstruct the non-Dalian part of recurrent E interactions, assuming that I neuron dynamics equilibrates faster than E neurons. Our work extends these previous accounts in two ways.

First, we take a more principled approach and propose a common minimax dynamics that E and I spiking neurons collectively optimize, adopting an idea from previous work on networks of rate-coding (continuous outputs) E-I neurons [16]. The minimax approach, besides other benefits, provides an intriguing interpretation of the antagonism between E and I neural populations.

Second, we consider minimax objectives that perform functions beyond signal reconstruction. Energy minimizing networks with symmetric interactions have been a powerful modeling framework for neuroscience since Hopfield's seminal contribution [14]. Phenomena such as associative memory [14], oculomotor integration [17,18], coding of periodic variables like head direction [19–23] and grating orientation [24], and spatial navigation [25,26] were modeled with fixed-point or manifold attractors resulting from such energy minimizing neural dynamics. Here, by moving from energy minimization to minimax optimization, we extend the reach of normative modeling to E-I spiking networks and provide derivations of circuits for each of the functions listed above.

Our technical contributions in this paper are:
- Derivation of a greedy spiking algorithm from a minimax objective. We discuss the optimality conditions leading to firing rate predictions of individual neurons and detailed balance. We argue that greedy spiking dynamics leads to tight balance, and provide necessary conditions for convergence of the dynamics.
- Applications. We design spiking networks that reconstruct signals, and exhibit fixed-point and manifold attractors, while obeying Dale's law and remaining in tight and detailed balance. We verify our theory in simulations. These applications indicate that our approach offers a principled method for designing spiking neural network models for cortical functions.

## 2   Minimax dynamics of optimally balanced E-I spiking networks

In this section, we consider the spiking dynamics of an integrate-and-fire E-I network, and show how such dynamics can be derived from a minimax objective function as a greedy optimization algorithm. By analyzing the optimum of the constrained minimax objective, we observe that the network is in detailed balance and derive conditions for stability of the optimum. Finally, we derive conditions for convergence of the dynamics and demonstrate tight balance. Our derivation extends the methods of [12,13] to minimax optima.

Our network is composed of separate E and I neuron populations of $N^E$ and $N^I$ neurons (Fig. 1a). For simplicity, only the E neurons receive $N^0$ dimensional external inputs, although our results and analysis still hold when both populations receive external inputs. The spiking dynamics is given by

$$
\frac{dV_i^E}{dt} = -\frac{V_i^E}{\tau_E} + \sum_j W_{ij}^{EE} s_j^E - \sum_j W_{ij}^{EI} s_j^I + \sum_j F_{ij} s_j^0,
$$
$$
\frac{dV_i^I}{dt} = -\frac{V_i^I}{\tau_I} + \sum_j W_{ij}^{IE} s_j^E - \sum_j W_{ij}^{II} s_j^I. \tag{1}
$$

Here, $V_i^E$, $i = 1, \ldots, N^E$ and $V_i^I$, $i = 1, \ldots, N^I$, denote the membrane potentials for E and I neurons respectively, $\tau_E$ and $\tau_I$ are the corresponding membrane time constants, and $s_j^E$ and $s_j^I$ denote the spike trains of E and I neurons: e.g. $s_j^{E(I)}(t) = \sum_k \delta(t - t_{j,k})$, where $t_{j,k}$ is the time of the $k$-th spike of the $j$-th neuron. $s_j^0$, $j = 1, \ldots, N^0$ denotes the input signal, which is not required to be a spike train. $\mathbf{F} \in \mathbb{R}^{N^E \times N^0}$ is the feed-forward connectivity matrix and $\mathbf{W}^{EE,EI,IE,II}$ are the connectivity matrices within and between the E-I populations. We require $\mathbf{W}^{EE} \in \mathbb{R}^{N^E \times N^E}$ and $\mathbf{W}^{II} \in \mathbb{R}^{N^I \times N^I}$ to be symmetric and $\mathbf{W}^{IE} = \mathbf{W}^{EI^\top}$ for our minimax objective approach. All the off-diagonal elements of the weight matrices are non-negative so that the network obeys Dale's law. The spiking reset is incorporated into the diagonals of $\mathbf{W}^{EE}$ and $\mathbf{W}^{II}$, which define how much the membrane potentials decrease after the arrival of a spike. Therefore, the diagonal elements of $\mathbf{W}^{EE}$ are negative and $\mathbf{W}^{II}$ are positive. The spiking thresholds for the E and I neurons are given by $T_i^E = -\frac{1}{2} W_{ii}^{EE}$ and $T_i^I = \frac{1}{2} W_{ii}^{II}$ respectively. We can obtain implicit expressions for $V_i^E$ and

$V_i^I$ by integrating Eq. (1):

$$V_i^E(t) = \sum_j W_{ij}^{EE} r_j^E(t) - \sum_j W_{ij}^{EI} r_j^I(t) + \sum_j F_{ij} x_j(t),$$

$$V_i^I(t) = \sum_j W_{ij}^{IE} r_j^E(t) - \sum_j W_{ij}^{II} r_j^I(t), \qquad (2)$$

where $r_j^{E,I}$ and $x_j$ are defined by filtering the spike train or the input signal with an exponential kernel:

$$r_j^{E,I}(t) = \int_0^\infty e^{-(t-t')/\tau_{E,I}} s_j^{E,I}(t-t')dt', \quad x_j(t) = \int_0^\infty e^{-(t-t')/\tau_E} s_j^0(t-t')dt'. \qquad (3)$$

Our intuition is that the network is in a balanced state when the dynamics reaches the optimum of an objective. Detailed balance of E and I inputs require that $V_i^{E,I}$ should fluctuate around zero for each neuron $i$ [6]. Tight balance requires the imbalance between E and I to be short-lived [6]. As we show next, both of these goals are achieved when the dynamics Eq. (1) performs a greedy optimization of a minimax objective $S$, where the implicit expressions Eq. (2) correspond to the saddle-point conditions.

**Spiking dynamics is a greedy algorithm optimizing a minimax objective:** Because I to E connections and E to I connections have opposite signs in Eq.s (1) and (2), a network that obeys Dale's law cannot be derived from a single minimizing objective. We instead consider the minimax optimization problem:

$$\min_{\mathbf{r}^E \geq 0} \max_{\mathbf{r}^I \geq 0} S(\mathbf{r}^E, \mathbf{r}^I), \quad S = -\frac{1}{2}\mathbf{r}^{E\top}\mathbf{W}^{EE}\mathbf{r}^E + \mathbf{r}^{E\top}\mathbf{W}^{EI}\mathbf{r}^I - \frac{1}{2}\mathbf{r}^{I\top}\mathbf{W}^{II}\mathbf{r}^I - \mathbf{x}^\top\mathbf{F}^\top\mathbf{r}^E. \qquad (4)$$

We can derive from our objective function a greedy algorithm that performs the optimization, which corresponds to the spiking dynamics Eq. (1), by adopting a similar approach to [12, 13]. The details of the derivation is provided in Supplementary Information (SI) A. Here we outline the approach. We track filtered spike trains, $r_i^{E,I}$ given in Eq. (3), instantaneously. At each time, each E/I neuron makes an individual decision to spike. An E neuron spikes if the spike decreases the instantaneous value of $S$: $S(\mathbf{r}^E + \mathbf{e}_i, \mathbf{r}^I) < S(\mathbf{r}^E, \mathbf{r}^I)$, where $\mathbf{e}_i$ is the $i$-th standard basis vector. An I neuron spikes if the spike increases the instantaneous value of $S$: $S(\mathbf{r}^E, \mathbf{r}^I + \mathbf{e}_i) > S(\mathbf{r}^E, \mathbf{r}^I)$. This series of spiking decisions lead to the spiking dynamics defined in Eq. (1) when $\tau_E = \tau_I = \tau$, and the dynamics in Eq.s (2) and (3) for the general case where the time constants can be different.

**KKT conditions imply detailed and tight balance of active neurons:** We can verify that the detailed balance is reached at the optimum of the constrained minimax problem by examining the KKT conditions [27] obtained by considering the optimizations with respect to $\mathbf{r}^I$ and $\mathbf{r}^E$ sequentially. These KKT conditions are satisfied for local optimal points of the minimax problem [28, 29]. We obtain, $\forall i \in \{1, \ldots, N^I\}$

$$r_i^I = 0, \quad \text{and} \quad V_i^I = \sum_j W_{ji}^{EI} r_j^E - \sum_j W_{ij}^{II} r_j^I \leq 0, \quad \text{and} \quad \lambda_i^I \geq 0,$$

$$\text{or} \quad r_i^I > 0, \quad \text{and} \quad V_i^I = \sum_j W_{ji}^{IE} r_j^E - \sum_j W_{ij}^{II} r_j^I = 0, \quad \text{and} \quad \lambda_i^I = 0, \qquad (5)$$

and $\forall i \in \{1, \ldots, N^E\}$

$$r_i^E = 0, \quad \text{and} \quad V_i^E = \sum_j W_{ij}^{EE} r_j^E - \sum_j W_{ij}^{EI} r_j^I + \sum_j F_{ij} x_j \leq 0, \quad \text{and} \quad \lambda_i^E \geq 0,$$

$$\text{or} \quad r_i^E > 0, \quad \text{and} \quad V_i^E = \sum_j W_{ij}^{EE} r_j^E - \sum_j W_{ij}^{EI} r_j^I + \sum_j F_{ij} x_j = 0, \quad \text{and} \quad \lambda_i^E = 0. \qquad (6)$$

From the KKT conditions, we see that for active neurons with $r_i^{E,I} > 0$, we have $V_i^{E,I} = 0$ at the saddle point, which suggests that any neuron that is active is in detailed balance.

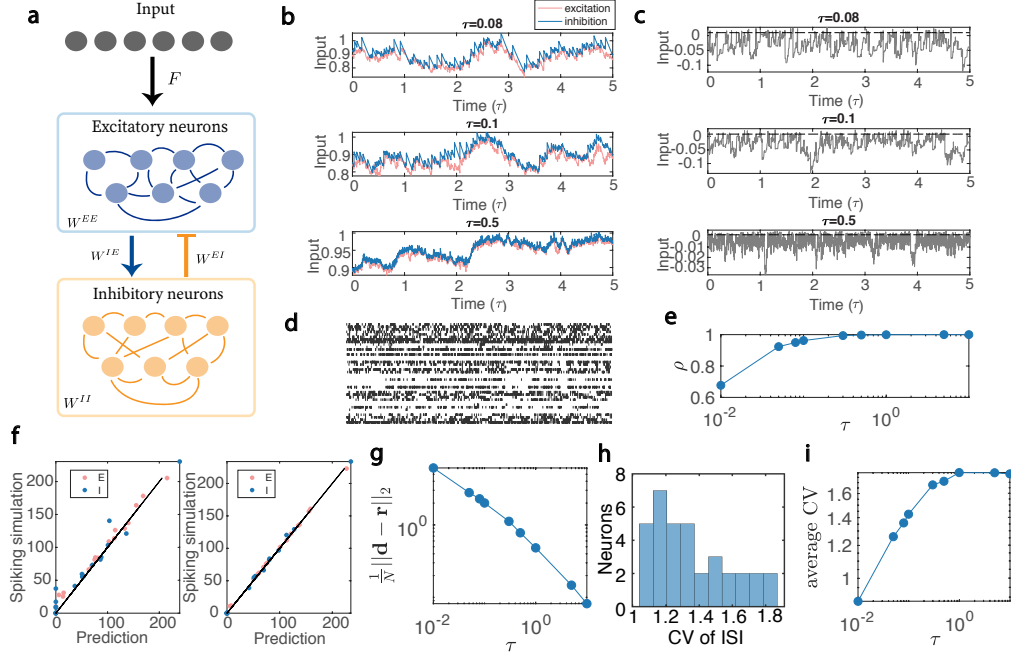

Figure 1: Optimally balanced E-I network. (a) Network of E-I neurons that obeys the Dale's law. Weights are designed to satisfy conditions in Eq.s (7) and (8). (b) E input ($\mathbf{W}^{EE}\mathbf{r}^E + \mathbf{Fx}$) and I input ($\mathbf{W}^{EI}\mathbf{r}^I$) for an active E neuron with different time constants $\tau$, normalized by the maximum value of the E input. (c) Net input ($\mathbf{W}^{EE}\mathbf{r}^E + \mathbf{Fx} - \mathbf{W}^{EI}\mathbf{r}^I$), normalized by the maximum value of the E input. Net input fluctuates around zero, occasionally going above the threshold (dashed line, also normalized). (d) Spiking pattern for an input with constant firing rate for $5\tau$, $\tau = 0.5$, corresponding to the same simulation time period as shown in the bottom panel of (b). (e) Pearson correlation coefficient between E and I input for an active neuron with different time constants $\tau$. The network becomes more tightly balanced with increasing $\tau$. (f) Prediction of firing rate for E and I populations. The prediction is obtained by directly solving the KKT conditions of the minimax optimization problem. Left: $\tau = 0.1$, Right: $\tau = 1$. Larger time constant results in more accurate prediction. (g) The error of firing rate prediction exhibits a power law dependence on the time constant. (h) Distribution of inter-spike interval (ISI) coefficient of variation (CV) for active neurons ($\tau = 0.08$). Most of the neurons have CV close to 1. (i) The average CV for active neurons increases as $\tau$ becomes larger, but remains close to 1, and saturates around 1.7.

Because of the greedy nature of optimization, when the network state deviates from the balanced state, the dynamics will automatically self-correct (imperfectly) by producing spikes when $V_i$ is larger than the spiking threshold $V_{th}$, leading to tight balance. For the inactive (non-spiking) neurons, KKT conditions state that the voltage $V_i < 0$. This is smaller than a positive $V_{th}$, consistent with the neuron never spiking.

We verify our conclusions with simulations. The E and I inputs for active neurons closely track each other (Fig. 1b), and their total input fluctuates around zero (Fig. 1c). Spike raster plot of the network is shown in Fig. 1d for $\tau = 0.5$. The network is more tightly balanced with increasing time constants $\tau$ as the Pearson correlation $\rho$ between E and I inputs for the same active neuron increases (Fig. 1e). We can predict individual neuron firing rates (defined as $f_i^{E,I}(t) = \frac{1}{\tau}r_i^{E,I}(t)$, a normalized version of $r_i^{E,I}$) in this spiking neural network by directly solving KKT conditions Eq.s (5) and (6) of the quadratic minimax problem Eq. (4). As shown in Fig. 1f, the prediction is quite accurate. We observe that the firing rate prediction becomes more accurate as $\tau$ increases (Fig. 1f&g). Similar observation was made for reconstruction error minimizing spiking networks [13]. Despite being tightly balanced, the spiking is also irregular. The coefficient of variation (CV) for active neurons are close to 1 (Fig. 1i), and the distribution of ISI is close to exponential (not shown). CV also increases as $\tau$ becomes larger, but it remains close to 1.

**Further optimality conditions:**    KKT conditions are necessary but not sufficient. The second order sufficient conditions [27–29] of the minimax problem (by considering maximization and minimization sequentially, details in SI B.1) give us

$$\hat{\mathbf{W}}^{II} \succcurlyeq 0, \qquad \hat{\mathbf{W}}^{EI}\hat{\mathbf{W}}^{II^{-1}}\hat{\mathbf{W}}^{IE} - \hat{\mathbf{W}}^{EE} \succcurlyeq 0, \tag{7}$$

where $\hat{\mathbf{W}}^{II}$ is the principal submatrix of $\mathbf{W}^{II}$ whose columns and rows correspond to nonzero elements of $\mathbf{r}^I$, $\hat{\mathbf{W}}^{EE}$ is the principal submatrix of $\mathbf{W}^{EE}$ whose columns and rows correspond to nonzero elements of $\mathbf{r}^E$, and $\hat{\mathbf{W}}^{EI}$ is a submatrix of $\mathbf{W}^{EI}$ with its rows corresponding to nonzero elements of $\mathbf{r}^E$ and its columns corresponding to nonzero elements of $\mathbf{r}^I$.

**Convergence conditions:**    The dynamics of approaching a saddle point of our minimax objective can be quite different than that of a minimum. The conditions discussed above describe an optimum of the objective, however, unlike in a minimization problem, where the objective itself act as Lyapunov function of the dynamics, these optimality conditions do not guarantee that the spiking dynamics converges to the optimum. The convergence of spiking dynamics is challenging to prove. Instead, we characterize the convergence conditions for a rate dynamics optimizing the minimax objective Eq. (4), hypothesizing that similar conditions would hold for the spiking dynamics. This thinking is also motivated by previous work that proved the convergence of a certain type of spiking dynamics to the fixed point of a rate dynamics, both minimizing the reconstruction error cost function [30, 31]. By constructing an energy (or Lyapunov) function [16, 32] on the rate dynamics, we derive the sufficient convergence conditions below (see SI B.2 for details):

1. The following eigenvalue condition guarantees the convergence of any bounded trajectory that does not exhibit a change in the set of active/inactive neurons through its evolution:

$$\lambda_{\min}(\hat{\mathbf{W}}^{II}) > \lambda_{max}(\hat{\mathbf{W}}^{EE}). \tag{8}$$

2. Replacing the positive semidefiniteness in the second order sufficient conditions for the optimality of the objective (Eq. (7)) with strict positive definiteness guarantees the boundedness of trajectories.

We empirically observed that these conditions are also valid for the convergence of spiking dynamics.

## 3    Applications

Having access to an objective function allows us to design balanced E-I networks to perform specific functions. Such normative approach has been common in neuroscience, using energy minimizing dynamics to derive efficient coding circuits [12, 15, 33–37]. Energy functions for various types of attractors have been thoroughly studied: fixed point attractor networks are commonly used as models for associative memory [14], ring attractor networks are used to model head-direction systems [19–23] and orientation selectivity [24] and grid attractor models are used to model grid cell responses [25, 26]. Here, we revisit all these systems and show how balanced E-I circuits performing the same tasks can be obtained by designing the minimax objectives and weights.

### 3.1    Input reconstruction

Consider the typical objective for a signal reconstruction problem: minimization of the mean squared error with an $l_2$-regularizer on the response,

$$\underset{\mathbf{r}^E}{\arg\min}\|\mathbf{x} - \mathbf{F}^\top\mathbf{r}^E\|_2^2 + \frac{\lambda}{2}\|\mathbf{r}^E\|_2^2 = \underset{\mathbf{r}^E}{\arg\min} -\mathbf{x}^\top\mathbf{F}^\top\mathbf{r}^E + \frac{1}{2}\mathbf{r}^{E^\top}\mathbf{F}\mathbf{F}^\top\mathbf{r}^E + \frac{\lambda}{2}\mathbf{r}^{E^\top}\mathbf{r}^E. \tag{9}$$

Our goal is to transform this problem to the one in Eq. (4), allowing a mapping to an E-I network. To achieve our goal, we first perform a matrix factorization $\mathbf{F}\mathbf{F}^\top = \mathbf{U}\mathbf{\Sigma}\mathbf{U}^\top$ where $\mathbf{U} \in \mathbb{R}^{N^E \times N^I}$, $\mathbf{\Sigma} \in \mathbb{R}_{\geq 0}^{N^I \times N^I}$. We emphasize that the elements of $\mathbf{\Sigma}$ are non-negative, $\mathbf{\Sigma}$ is symmetric and can be, but does not have to be, diagonal. For the factorization to be plausible $N^I \geq \text{rank}(\mathbf{F})$. We will show examples of two different ways of performing the factorization, 1) by simply choosing $\mathbf{\Sigma} = \mathbf{I}$ and $\mathbf{U} = \mathbf{F}$, and 2) performing a singular value decomposition (SVD) on $\mathbf{F}\mathbf{F}^\top$. We also separate out the positive and negative parts in $\mathbf{U}$, and denote them by $[\mathbf{U}]_+ = \mathbf{U}_+$, and $[\mathbf{U}]_- = \mathbf{U}_-$. With all this set

up, the reconstruction error cost function can be written as:

$$\min_{\mathbf{r}^E} -\mathbf{x}^\top \mathbf{F}^\top \mathbf{r}^E + \mathbf{r}^{E\top}\left(\mathbf{U}_+\boldsymbol{\Sigma}\mathbf{U}_-^\top + \mathbf{U}_-\boldsymbol{\Sigma}\mathbf{U}_+^\top + \frac{\lambda}{2}\mathbf{I}\right)\mathbf{r}^E + \frac{1}{2}\mathbf{r}^{E\top}\left(\mathbf{U}_+ - \mathbf{U}_-\right)\boldsymbol{\Sigma}\left(\mathbf{U}_+ - \mathbf{U}_-\right)^\top \mathbf{r}^E.$$
(10)

Next, using a "variable substitution" trick commonly used in statistical mechanics and the similarity matching framework [38], we transform the problem into a minimax problem and introduce the I neurons:

$$\min_{\mathbf{r}^E}\max_{\mathbf{r}^I} -\mathbf{x}^\top \mathbf{F}^\top \mathbf{r}^E + \mathbf{r}^{E\top}\left(\mathbf{U}_+\boldsymbol{\Sigma}\mathbf{U}_-^\top + \mathbf{U}_-\boldsymbol{\Sigma}\mathbf{U}_+^\top + \frac{\lambda}{2}\mathbf{I}\right)\mathbf{r}^E + \mathbf{r}^{I\top}\boldsymbol{\Sigma}\left(\mathbf{U}_+ - \mathbf{U}_-\right)\mathbf{r}^E - \frac{1}{2}\mathbf{r}^{I\top}\boldsymbol{\Sigma}\mathbf{r}^I.$$
(11)

The equivalence of Eq. (11) to Eq. (10) can be seen by performing the $\mathbf{r}^I$ maximization explicitly. With this formulation, we have $\mathbf{W}^{EE} = 2(\mathbf{U}_+\boldsymbol{\Sigma}\mathbf{U}_-^\top + \mathbf{U}_-\boldsymbol{\Sigma}\mathbf{U}_+^\top + \frac{\lambda}{2}\mathbf{I})$, $\mathbf{W}^{IE} = \boldsymbol{\Sigma}(\mathbf{U}_+ - \mathbf{U}_-)$, and $\mathbf{W}^{II} = \boldsymbol{\Sigma}$. Next, we present simulations of this network.

**Reconstruction of synthetic data**   First, we test our network on a synthetic dataset with randomly generated $N^0$ dimensional i.i.d. uniformly distributed input in $[0, r_{max}]^{N^0}$. We choose $\boldsymbol{\Sigma} = \mathbf{I}$, i.e. there are no recurrent connections between I neurons. To be able to reconstruct all non-negative inputs, we choose $\mathbf{F}$ such that $\mathbf{F}^+ = \mathbf{F}(\mathbf{F}^\top\mathbf{F})^{-1}$ is non-negative. We then design our weights correspondingly to satisfy the constraints described in Eq.s (7) and (8). In Fig. 2a, we see that the reconstruction is remarkably accurate. In this example dimensionality is expanded, i.e. $N^E > N^0$, but $N^I = \mathrm{rank}(\mathbf{F}) = \mathrm{N}^0$ to satisfy the factorization rank constraints.

We also simulate time varying inputs. Here, the network computes a leaky integration of the input signal dependent on the timescale of the network. We simulate step inputs given by $s(t)$, as shown in Fig. 2b. The target line is obtained by directly computing $\frac{dx}{dt} = -\frac{x}{\tau} + s(t)$, where $\tau = \tau_E = \tau_I$. The prediction line is given by decoding the firing rate of the spiking network, i.e. computing $\mathbf{F}^\top\mathbf{r}^E(t)$.

**Reconstruction of natural image patches with varying number of I neurons.**   Next, we reconstruct natural image patches. Using the sparse coding algorithm of [33], we learn a dictionary for $13 \times 13$ natural image patches, concatenated to the rows of the matrix $\mathbf{F}$. We build our network using an SVD, $\mathbf{F}\mathbf{F}^\top = \mathbf{U}\boldsymbol{\Sigma}\mathbf{U}^\top$, and keep only the largest $N^I$ singular values. For this example we allow $N^I < \mathrm{rank}(\mathbf{F})$, leading to signal loss. We observe that the reconstruction becomes more accurate as we increase the number of I neurons (Fig. 2c&d) because larger ratio of the variance of the dictionary we learned from natural image patches is captured by the I neurons (Fig. 2e).

## 3.2   Attractor networks

Attractor networks are widely used as models of neural and cognitive phenomena [14, 19–22, 24, 26, 39, 40]. Frequently such attractor networks minimize energy functions and violate Dale's law. Here, we show how minimax objectives can be used to design optimally balanced E-I attractor networks for popular manifold attractors used in neuroscience.

**Fixed point attractors.**   Fixed point attractors have been used to model associative memory [14]. We can store attractors in our network as discrete fixed points with properly designed weights, allowing different fixed points to have overlapping sets of active neurons. In Fig. 3, for a given set of fixed points, the network weights are obtained by performing a nonlinear constrained optimization problem. We minimize the mean-squared-error of the membrane potentials $V_i^{E,I}$ for the active neurons at the fixed points, subject to the constraints given by the KKT conditions for the inactive neurons (Eq.s (5) and (6)), the second order sufficient (Eq. (7)) condition, and the condition for convergence of rate dynamics (Eq. (8)) (see SI C.1). We show in Fig. 3 that with the optimized weights, our network can store different attractors (Fig. 3a&b) while remaining tightly balanced (Fig. 3c).

**Ring attractor.**   Ring attractors have been used to model the head direction system [19–23] and orientation selectivity in primary visual cortex [24]. Its dynamics can be derived from energy functions. To store a ring attractor in our E-I network, we design the weight matrices such that the

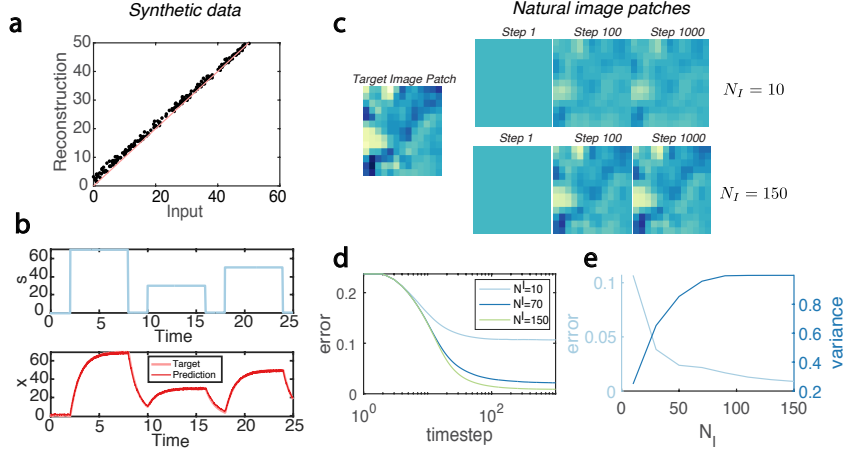

Figure 2: Reconstruction of input signal: (a) Our network with $N^E = 60, N^I = 10$ achieves input reconstruction accurately. The input is 20 different 10-dimensional i.i.d. signals sampled from a uniform distribution with $r_{max} = 50$. (b) Reconstruction of filtered step inputs can be achieved with our network. $\tau = 1$. (c) Reconstruction of a target image patch at simulation steps 1, 100 and 1000 for $N^E = 400$ and $N^I = 10$ and $N^I = 150$. (d) Reconstruction error $\frac{1}{N^0}\|\mathbf{F}^T \mathbf{r}^E - \mathbf{x}\|^2$ measured as a function of time, for the $N^I = 10, 70, 150$. (e) Left: The error after convergence measured as a function of $N_I$. Right: The ratio of variance that is accounted for by the $N_I$ neurons.

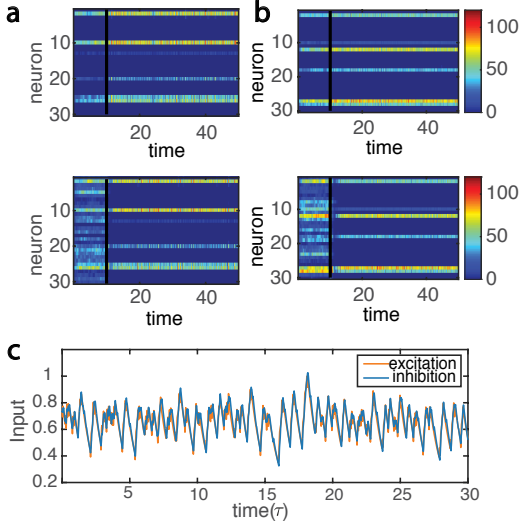

Figure 3: Balanced E-I network trained to store 2 attractor states with $N^E = N^I = 15$: (a),(b) Two different attractor states with overlapping sets of active neurons in the same spiking E-I network, the network is initialized with different inputs that stop after 10s, the network correctly converges to the more adjacent fixed point afterwards. The simulation timescales are $\tau_E = 0.5, \tau_I = 0.4$. Top: noiseless case, Bottom: initialized with noisy inputs. Color indicates the firing rate. (c) The E input and I input for an active neuron (normalized by maximum E input) closely track each other. The network remains tightly balanced in the attractor states.

effective energy function for the E neurons matches the energy function in a standard ring attractor network [20]:

$$\mathcal{L} = -\frac{1}{2N^E}\sum_{ij} r_i^E (w_0 + w_1\cos(\theta_i - \theta_j))r_j^E - \sum_i (h_0 + h_1\cos(\theta_0 - \theta_i))r_i^E + \sum_i \int_0^{r_i^E} f^{-1}(r_i^E),$$

(12)

where $w_0, w_1, h_0, h_1$ are scalar parameters that control the system's behavior (see SI C.2), and $f(x) = [x]_+$ represents ReLU nonlinearity (see SI B.2). Each neuron $i$ in the E population is assigned a distinct preferred angle $\theta_i \in \left\{0, \frac{2\pi}{N^E}, \dots, \frac{2\pi(N^E - 1)}{N^E}\right\}$.

We assume that the inhibitory neurons are all active ($r_i^I(t) > 0$) at the optima for simplicity, which can be achieved by carefully designing the weights (eg. $\mathbf{W}^{II^{-1}}\mathbf{W}^{IE} > 0$). We obtain an effective energy function for E neurons by maximizing over $\mathbf{r}^I$ in Eq. (4) and plugging in the optimum value:

$$L_{eff} = -\frac{1}{2}\mathbf{r}^{E^\top}(\mathbf{W}^{EE} - \mathbf{W}^{EI}\mathbf{W}^{II^{-1}}\mathbf{W}^{IE})\mathbf{r}^E - \mathbf{x}^\top \mathbf{F}^\top \mathbf{r}^E.$$

(13)

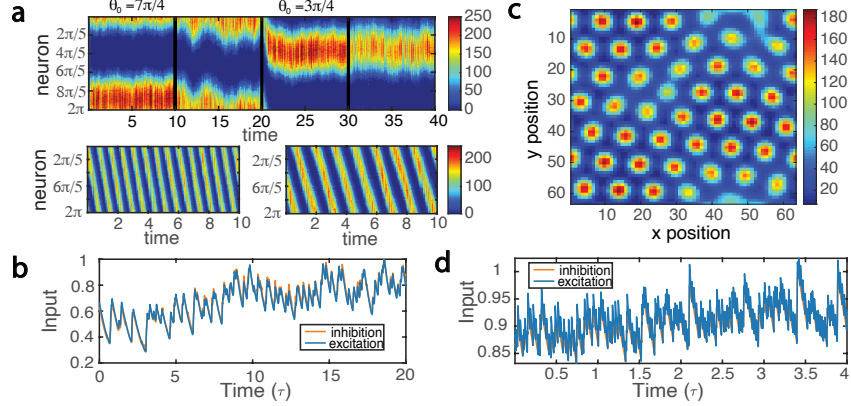

Figure 4: Ring attractor and grid attractor in E-I balanced network: (a) Top: During 0-10s and 20-30s, the network is initialized with inhomogeneous input with two different $\theta_0$'s, this inhomogeneity is turned off during 10-20s and 30-40s. The network is able to pick up inhomogeneous response even when the input to each neuron is identical, the attractor state the network ends up in depends on initialization. Bottom: Spiking network with the same parameters as in corresponding panel in (a), except that $h_1$ is zero throughout the simulation (Left: $\gamma = 0.15$, Right: $\gamma = 0.08$). Larger $\gamma$ corresponds to higher angular frequency. (b) The network remains tightly balanced in the attractor states. The E and I inputs for $20\tau$ ($\tau = 0.1$) are both normalized by the maximum of the E input, they closely track each other. (c) Spiking model for grid attractors. Each pixel represents a neuron. Color indicates firing rate of individual neurons as defined in Eq. (3). (d) E and I input for one of the active neurons in (c) for a duration of $4\tau_E$, both normalized by the maximum of the E input. The network is tightly balanced.

To match this effective energy function with Eq. (12), we design weights such that $(\mathbf{W}^{EE} - \mathbf{W}^{EI}{\mathbf{W}^{II}}^{-1}\mathbf{W}^{IE})_{ij} = \frac{1}{N_E}(w_0 + w_1\cos(\theta_i - \theta_j)) - \delta_{ij}$ ($\delta_{ij}$ term is due to the nonlinearity, similar to SI B.2). For simplicity, we choose $\mathbf{W}^{II}$ to be an identity matrix and $\mathbf{W}^{EI}$ to be a uniform matrix whose elements are all equivalent. The amplitude of the two matrices are tuned for the constraints to be satisfied. The input to the $i$-th neuron is given by $h_0 + h_1\cos(\theta_0 - \theta_i)$. $\theta_0$ reflects the inhomogeneity of the input: if $h_1 > 0$, neurons with their preferred angle $\theta_i$ closer to $\theta_0$ receive larger inputs. As the standard ring attractor model, this network exhibits different behaviors in different parameter regimes [41] (see SI C.2). In a certain regime the network self-organizes into a bump attractor state even with homogeneous input (Fig. 4a). It also exhibits tight balance (Fig. 4b).

We can introduce anisotropy to our weights to obtain further functions while loosing the objective function interpretation. By following [41, 42], when we design the weights as $(\mathbf{W}_{EE} - \mathbf{W}_{EI}(\mathbf{W}_{II})^{-1}\mathbf{W}_{IE})_{ij} = \frac{1}{N_E}(w_0 + w_1\cos(\theta_i - \theta_j) - w_1\gamma\sin(\theta_i - \theta_j)) - \delta_{ij}$ ($\mathbf{W}^{II}$ and $\mathbf{W}^{EI}$ are chosen as above), the network produces traveling waves, where $\gamma$ controls the angular frequency (Fig. 4a). In this example, the network stays on a limit cycle and does not reach an equilibrium, and therefore is not balanced (SI C.2).

**Grid attractor.** We next discuss a grid attractor, which is used to model the grid cells of the entorhinal cortex [26]. Here, neurons are arranged on a square sheet with periodic boundary conditions (a torus). Each neuron is assigned a positional index $\mathbf{x}_i$. We design the weight matrices such that the effective energy function for the E neurons resembles that of the grid attractor :

$$L = -\frac{1}{2}\sum_{ij} r_i^E W_0(\mathbf{x}_i - \mathbf{x}_j)r_j^E + \sum_i A_i r_i^E + \sum_i \int_0^{r_i^E} f^{-1}(r_i^E), \qquad (14)$$

where $W_0(\mathbf{x}) = ae^{-\gamma|\mathbf{x}|^2} - e^{-\beta|\mathbf{x}|^2}$ [26]. We design the weights such that $(\mathbf{W}_{EE} - \mathbf{W}_{EI}(\mathbf{W}_{II})^{-1}\mathbf{W}_{IE})_{ij} = W_0(\mathbf{x}_i - \mathbf{x}_j) - \delta_{ij}$ ($\mathbf{W}^{II}$ and $\mathbf{W}^{EI}$ are chosen same as the ring attractor, for detailed parameters see SI C.3). Neurons receive homogeneous input, i.e. $A_i = A, \forall i$. We see that although the input is homogeneous, the spiking neural network self-organizes into a grid-like

pattern reminiscent of grid cells on the entorhinal cortex (Fig. 4c). Furthermore, the active neurons in the network remain tightly balanced with the E-I currents cancelling each other precisely (Fig. 4d).

## 4 Conclusion

Our work provides a novel normative framework for modeling of biologically realistic spiking neural networks. While energy minimization has been used widely in computational neuroscience [12, 14, 15, 19–26, 33–37], we extend normative modeling to minimax optimization in optimally balanced E-I spiking networks. Compared to existing work on minimax dynamics of rate neurons [16], we made several major contributions. We extended the framework to spiking neurons and by inspecting the KKT conditions, we proved that the active neurons are in detailed balance at the saddle point. Simulations confirmed tight balance. We also described conditions on the weights for the rate dynamics (gradient descent-ascent) to converge. Furthermore, by relating our effective objective to the well-studied energy functions for different types of computational goals, we designed novel E-I circuit architectures for various applications from minimax objectives, where the saddle points can be either discrete or forming manifolds, similar to the minima in traditional minimization objective formulations. These applications could be potentially useful for the neuroscience community. Finally, minimax objectives can be a useful strategy when faced with uncertainty and previous works have demonstrated that minimax objectives can be used to solve problems such as source localization [43] and sensorimotor control [44].

## Broader Impact

This work introduces a principled approach to designing spiking neural networks of excitatory (E) and inhibitory (I) neurons to perform various computations. As any spiking neural network model, networks derived from our approach could be applied in the field of neuromorphic computing, where information is transmitted through spikes instead of rate, and is thus more energy efficient. Previous work have shown that E-I balanced networks can serve as fast responding modules [9, 10], and our approach could be applied to designing E-I balanced modules that potentially speed up solving optimization problems in general neuromorphic computing systems.

Furthermore, we provided several conditions for the regular functioning of our spiking networks. These conditions could potentially have implications on understanding neural connectivity in the cortex, and how pathological activities in the brain may arise from disrupted synaptic interactions.

## Acknowledgments and Disclosure of Funding

We thank Sreya Vemuri for help in the initial stages of this project. This work was supported by funds from the Intel Corporation.

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
