[Supplementary Material]

# Supplementary Information (SI)

## A  Spiking dynamics as a greedy optimization algorithm on the minimax objective

We first show the derivation of the spiking dynamics for I neurons. Because we are maximizing over $\mathbf{r}^I$, the neuron fires a spike when it increases the objective. The firing condition for neuron $k$ correspond to $S(\mathbf{r}^E, \mathbf{r}^I + \mathbf{e}_k) > S(\mathbf{r}^E, \mathbf{r}^I)$, where $\mathbf{e}_k$ denotes the standard basis vector. By plugging in Eq. (4), we have

$$\sum_{ij} r_i^E W_{ij}^{EI} r_j^I - \frac{1}{2} \sum_{ij} r_i^I W_{ij}^{II} r_j^I < \sum_{ij} (r_i^E W_{ij}^{EI}(r_j^I + \delta_{jk}) - \frac{1}{2}(r_i^I + \delta_{ik}) W_{ij}^{II}(r_j^I + \delta_{jk}))$$
$$\implies \sum_i W_{ik}^{EI} r_i^E - \sum_j r_j^I W_{jk}^{II} > \frac{1}{2} W_{kk}^{II}, \tag{SI.1}$$

where $\delta_{jk}$ denotes the $j$-th element of the standard basis vector $\mathbf{e}_k$. We define the membrane potential and the firing threshold of the I neurons.

$$V_k^I \equiv \sum_i W_{ik}^{EI} r_i^E - \sum_i W_{ik}^{II} r_i^I$$
$$T_k^I \equiv \frac{1}{2} W_{kk}^{II}, \tag{SI.2}$$

Next, we derive the dynamics of the membrane potential. For $\tau_E = \tau_I$ and using Eq. (3) for the definition of $r_i^{I/E}$, we arrive at:

$$\frac{dV_k^I}{dt} = -\frac{1}{\tau} V_k^I - \sum_i W_{ik}^{II} s_i^I + \sum_i W_{ik}^{EI} s_i^E. \tag{SI.3}$$

We recognize this as the standard integrate-and-fire spiking dynamics with threshold $T_k^I = \frac{1}{2} W_{kk}^{II}$ [45].

For the E neurons, we proceed similarly. The firing condition for a neuron $k$ is $S(\mathbf{r}^E + \mathbf{e}_i, \mathbf{r}^I) < S(\mathbf{r}^E, \mathbf{r}^I)$, where $\mathbf{e}_i$ is the $i$-th standard basis vector. Eq. (4) implies

$$\sum_{ij} -\frac{1}{2} r_i^E W_{ij}^{EE} r_j^E + r_i^E W_{ij}^{EI} r_j^I - \sum_{ij} x_i F_{ij} r_j^E$$
$$> \sum_{ij} (-\frac{1}{2}(r_i^E + \delta_{ik}) W_{ij}^{EE}(r_j^E + \delta_{jk}) + (r_i^E + \delta_{ik}) W_{ij}^{EI} r_j^I - \sum_{ij} x_i F_{ij}(r_j^E + \delta_{jk})) \tag{SI.4}$$
$$\implies \sum_i (W_{ki}^{EE} r_i^E - W_{ki}^{EI} r_i^I + F_{ik} x_i) > -\frac{1}{2} W_{kk}^{EE}.$$

Defining the membrane potential and the firing threshold

$$V_k^E \equiv \sum_i W_{ki}^{EE} r_i^E - \sum_i W_{ki}^{EI} r_i^I + \sum_i F_{ik} x_i$$
$$T_k^I \equiv -\frac{1}{2} W_{kk}^{EE}, \tag{SI.5}$$

we can obtain dynamics of the membrane potential for $\tau_E = \tau_I = \tau$:

$$\frac{dV_k^E}{dt} = -\frac{1}{\tau} V_k^E + \sum_i W_{ki}^{EE} s_i^E - \sum_i W_{ki}^{EI} s_i^I + \sum_i F_{ik} s_i^0, \tag{SI.6}$$

with spiking threshold $T_k^E = -\frac{1}{2} W_{kk}^{EE}$.

## B  Convergence of the dynamics

### B.1  Second order sufficient condition for optimality

We cite a theorem from [46].

**Theorem 1** *The solution $x^*, \lambda^*$ obeying the KKT conditions is a constrained local minimum if for the Lagrangian*

$$L(x, \lambda) = f(x) + \sum_{i=1}^m \lambda_i g_i(x), \tag{SI.7}$$

*we have*

$$\mathbf{s}^T \nabla^2_{xx} L(x^*, \lambda^*) \mathbf{s} \geq 0, \qquad (\text{SI.8})$$

*where $\mathbf{s} \neq 0$ is a vector satisfying*

$$\nabla_x g_i(x^*)^T \mathbf{s} = 0, \qquad (\text{SI.9})$$

*where only those active inequality constraints $g_i(x)$ corresponding to strict complimentarity (i.e. where $\lambda_i > 0$) are applied.*

We apply Thm. 1 to our minimax objective, for the maximization problem with $\mathbf{s}$ as the activities of inhibitory neurons that are active, and $\nabla_{xx} L(x^*, \lambda^*)$ gives $\hat{\mathbf{W}}^{II}$. $\hat{\mathbf{W}}^{II}$ is the submatrix where strict complimentarity is applied (namely when $\mathbf{r}^I > 0$), and therefore the second order sufficient condition for optimality is $\hat{\mathbf{W}}^{II} \succcurlyeq 0$. Plugging in the optimal solution for $\mathbf{r}^I$ in the maximization problem and similarly deriving the condition for optimality for the minimization over $\mathbf{r}^E$ according to Thm. 1, we obtain $\hat{\mathbf{W}}^{EI} \hat{\mathbf{W}}^{II^{-1}} \hat{\mathbf{W}}^{IE} - \hat{\mathbf{W}}^{EE} \succcurlyeq 0$.

## B.2 Rate dynamics and convergence

We can prove the convergence of a rate dynamics derived from the same minimax objective, for trajectories that do not include switching between active and inactive neurons, i.e., active neurons at initialization remain active, and silent neurons remain silent throughout the trajectory. In practice we observe convergence even when a transition between active and inactive states occurs during the dynamics for some cases.

We use a slightly modified objective function of the form

$$S = -\frac{1}{2} \mathbf{r}^{E^T} \mathbf{W}^{EE^*} \mathbf{r}^E + \mathbf{r}^{E^T} \mathbf{W}^{EI} \mathbf{r}^I - \frac{1}{2} \mathbf{r}^{I^T} \mathbf{W}^{II^*} \mathbf{r}^I - \mathbf{x}^T \mathbf{F} \mathbf{r}^E + \sum_i H(r_i^E) - \sum_i G(r_i^I). \quad (\text{SI.10})$$

The last two terms are related to nonlinear neural activations. For ReLU neurons with thresholds $\theta_E$ and $\theta_I$, $H(r_i^E) = \int_0^{r_i^E} f^{-1}(x)dx = \frac{1}{2}(r_i^E + \theta_i^E)^2$, $G(r_i^I) = \int_0^{r_i^I} f^{-1}(x)dx = \frac{1}{2}(r_i^I + \theta_i^I)^2$. Function $H(x)$ and $G(x)$ are only defined for $x \geq 0$. If we choose $\mathbf{W}^{EE^*} = \mathbf{W}^{EE} + \mathbf{I}$, $\mathbf{W}^{II^*} = \mathbf{W}^{II} - \mathbf{I}$, $\theta_E = \theta_I = 0$, then this objective reduces to Eq. (4). For the modifications induced by nonzero thresholds, one can simply change the thresholds of E neurons from $T_i^E = -\frac{1}{2} W_{ii}^{EE}$ to $T_i^E = -\frac{1}{2} W_{ii}^{EE} + \theta_i^E$, and that of the I neurons from $T_i^I = \frac{1}{2} W_{ii}^{II}$ to $T_i^I = \frac{1}{2} W_{ii}^{II} + \theta_i^I$.

The rate dynamics that optimizes this objective is

$$\tau_I \frac{du_i^I}{dt} = -u_i^I + \sum_j W_{ij}^{IE} r_j^E - \sum_j (W_{ij}^{II} - \delta_{ij}) r_j^I$$

$$\tau_E \frac{du_i^E}{dt} = -u_i^E + \sum_j (W_{ij}^{EE} + \delta_{ij}) r_j^E - \sum_j W_{ij}^{EI} r_j^I + \sum_j F_{ij}^T x_j \qquad (\text{SI.11})$$

$$r_j^{E(I)} = f(u_j^{E(I)}) = [u_j^{E(I)} - \theta_j^{E(I)}]_+.$$

To investigate convergence, we can construct an energy function of the system [16] assuming $\tau_E = \tau_I = \tau$:

$$L = \frac{1}{2} |\dot{\mathbf{r}^E}|^2 + \frac{1}{2} |\dot{\mathbf{r}^I}|^2 + \gamma S, \qquad \gamma \in \mathbb{R}. \qquad (\text{SI.12})$$

Next we show that the energy function is decreasing. Except for a set of measure zero ($\forall i, u_i^{E/I} = \theta_i^{E/I}$), the ReLU function is differentiable and we have

$$\dot{r_i}^{E/I} = \frac{dr_i^{E/I}}{du_i^{E/I}} \dot{u_i}^{E/I}, \qquad \ddot{r_i}^{E/I} = \frac{d^2 r_i^{E/I}}{du_i^{E/I^2}} \dot{u_i}^{E/I^2} + \frac{dr_i^{E/I}}{du_i^{E/I}} \ddot{u_i}^{E/I} = \frac{dr_i^{E/I}}{du_i^{E/I}} \ddot{u_i}^{E/I}. \qquad (\text{SI.13})$$

We compute the time derivative of the energy function:

$$\dot{L} = (\dot{\mathbf{r}}^E, \dot{\mathbf{r}}^I)^T (\ddot{\mathbf{r}}^E, \ddot{\mathbf{r}}^I) + \gamma \dot{S}$$

$$= (\dot{\mathbf{u}}^E, \dot{\mathbf{u}}^I)^T \begin{bmatrix} \frac{d\mathbf{r}^E}{d\mathbf{u}^E} & 0 \\ 0 & \frac{d\mathbf{r}^I}{d\mathbf{u}^I} \end{bmatrix} \begin{bmatrix} -\mathbf{I} + \mathbf{W}^{EE^*} \frac{d\mathbf{r}^E}{d\mathbf{u}^E} & -\mathbf{W}^{EI} \frac{d\mathbf{r}^I}{d\mathbf{u}^I} \\ \mathbf{W}^{IE} \frac{d\mathbf{r}^E}{d\mathbf{u}^E} & -\mathbf{I} - \mathbf{W}^{II^*} \frac{d\mathbf{r}^I}{\mathbf{u}^I} \end{bmatrix} (\dot{\mathbf{u}}^E, \dot{\mathbf{u}}^I)$$

$$+ \gamma(-\dot{\mathbf{u}}^E, \dot{\mathbf{u}}^I)^T \begin{bmatrix} \frac{d\mathbf{r}^E}{d\mathbf{u}^E} & 0 \\ 0 & \frac{d\mathbf{r}^I}{d\mathbf{u}^I} \end{bmatrix} (\dot{\mathbf{u}}^E, \dot{\mathbf{u}}^I) \qquad (\text{SI.14})$$

$$= (\dot{\mathbf{u}}^E, \dot{\mathbf{u}}^I)^T \begin{bmatrix} \frac{d\mathbf{r}^E}{d\mathbf{u}^E} \mathbf{W}^{EE^*} \frac{d\mathbf{r}^E}{d\mathbf{u}^E} - \frac{d\mathbf{r}^E}{d\mathbf{u}^E} - \gamma \frac{d\mathbf{r}^E}{d\mathbf{u}^E} & 0 \\ 0 & \frac{d\mathbf{r}^I}{d\mathbf{u}^I} \mathbf{W}^{II^*} \frac{d\mathbf{r}^I}{d\mathbf{u}^I} - \frac{d\mathbf{r}^I}{d\mathbf{u}^I} + \gamma \frac{d\mathbf{r}^I}{d\mathbf{u}^I} \end{bmatrix} (\dot{\mathbf{u}}^E, \dot{\mathbf{u}}^I)$$

$$\dot{L} = \dot{\hat{\mathbf{u}}}^{E^T} (\hat{\mathbf{W}}^{EE} - \gamma \mathbf{I}) \dot{\hat{\mathbf{u}}}^E - \dot{\hat{\mathbf{u}}}^{I^T} (\hat{\mathbf{W}}^{II} - \gamma \mathbf{I}) \dot{\hat{\mathbf{u}}}^I.$$

If $\hat{\mathbf{W}}^{II} - \gamma\mathbf{I}$ is positive definite and $\hat{\mathbf{W}}^{EE} - \gamma\mathbf{I}$ is negative definite, then any bounded trajectory that does not cross $u_i = 0$ boundaries (and cause changes in the set of active/inactive neurons) are convergent. If $\min\lambda(\hat{\mathbf{W}}^{II}) > \max\lambda(\hat{\mathbf{W}}^{EE})$, then there exist $\gamma$ such that $\min\lambda(\hat{\mathbf{W}}^{II}) > \gamma > \max\lambda(\hat{\mathbf{W}}^{EE})$, and the condition is satisfied.

Now we proved that any bounded trajectories that do not cross the $u_i = 0$ boundaries are convergent, we need to further prove that the trajectories are bounded. We applying Thm. 2 in [32].

**Theorem 2** *Given a twice differentiable objective S, suppose that $\lambda_{inf}(S_{\mathbf{xx}}) > \lambda_{sup}(S_{\mathbf{yy}})$. If either*

1. *$\lambda_{inf}(S_{\mathbf{xx}} > 0)$ and $-V(\mathbf{y}) = -\min_{\mathbf{x}} S(\mathbf{x},\mathbf{y})$ is radially unbounded, or*

2. *$\lambda_{inf}(S_{\mathbf{yy}} < 0)$ and $U(\mathbf{y}) = \max_{\mathbf{y}} S(\mathbf{x},\mathbf{y})$ is radially unbounded*

*is also satisfied, then any trajectory of gradient descent-ascent is bounded.*

We see that the condition for boundedness is the same as the condition for second order optimality as discussed in SI. B.1.

## C  Attractor networks

### C.1  Optimization problem for designing weights in fixed point attractor networks

We design the network weights in order to store fixed point attractors by solving an optimization problem. For a given attractor state $m$, we denote the set of active neurons as $\mathcal{A}_m$ and the set of silent neurons as $\mathcal{I}_m$. We formulate the following optimization problem.

$$\min \sum_m \sum_{i\in\mathcal{A}_m} \|V_i^m\|^2, \qquad \text{subject to}$$

$$\exists \kappa > 0,\ V_i^m \leq -\kappa,\ \forall i \in \mathcal{I}_m,\ \forall m \qquad\qquad \text{(constraint 1)}$$

$$\hat{\mathbf{W}}_m^{EI}\hat{\mathbf{W}}_m^{II^{-1}}\hat{\mathbf{W}}_m^{IE} - \hat{\mathbf{W}}_m^{EE} \succcurlyeq 0,\ \forall m \qquad\qquad \text{(constraint 2)}$$

$$\exists \gamma_m \in \mathbb{R},\ \min\lambda(\hat{\mathbf{W}}_m^{II}) > \gamma_m > \max\lambda(\hat{\mathbf{W}}_m^{EE}),\ \forall m \qquad \text{(constraint 3)}$$

$$\mathbf{W}^{EI/II/IE} \geq 0 \qquad\qquad \text{(constraint 4)}$$

$$W_{ij}^{EE} \geq 0,\ \forall i \neq j;\ W_{ii}^{EE} \leq 0,\ \forall i \qquad\qquad \text{(constraint 5)}$$

$$\mathbf{W}^{EE} = \mathbf{W}^{EE^T};\ \mathbf{W}^{II} = \mathbf{W}^{II^T};\ \mathbf{W}^{EI} = \mathbf{W}^{IE^T} \qquad \text{(constraint 6).} \qquad \text{(SI.15)}$$

Here $\hat{\mathbf{W}}_m^{EI/EE/IE/II}$ denote the submatrices with rows and columns corresponding to the active neurons in attractor state $m$. The expression for $V_i^m$ is given by plugging in $\mathbf{r}^E$ and $\mathbf{r}^I$ in the attractor state $m$ into Eq. (2) for E and I neurons respectively. Constraint 1 corresponds to the KKT condition on the inactive neurons. Constraints 2 and 3 guarantee convergence of rate dynamics when the trajectory does not cross the $u_i = 0$ boundaries. Constraints 4,5,6 are imposed for symmetry and nonnegativity of the matrices.

This is a nontrivial optimization problem with nonlinear constraints. We used the Sequential Quadratic Programming (SQP) algorithm for MATLAB function *fmincon* [47] to solve this optimization problem. We start with different initial values and only accept solutions that satisfy $\min\sum_m \sum_{i\in\mathcal{A}_m}\|V_i^m\|^2 = 0$.

### C.2  Different parameter regimes in ring attractor network

The energy function of a standard ring attractor model is given by

$$L = -\frac{1}{2N_E}\sum_{ij} r_i^E(w_0 + w_1\cos(\theta_i - \theta_j))r_j^E - \sum_i (h_0 + h_1\cos(\theta_0 - \theta_i))r_i^E + \sum_i \int_0^{r_i^E} f^{-1}(r_i^E) \quad \text{(SI.16)}$$

Here $h_0 + h_1\cos(\theta_0 - \theta_i)$ is the input term, and $f(x) = [x]_+$ is ReLU nonlinearity. This network has different parameter regimes for different behaviors [41]. When $w_0 \geq 1$, the network is unstable and the firing rate goes to infinity. When $w_1 < 2$, we have signal amplification, the network has larger response amplitude than its input. Finally, when $w_1 \geq 2$, there is symmetry breaking, the network can pick up inhomogeneous response even when the input is homogeneous ($h_1 = 0$). The stable states of the network lie on a ring, and which steady state the network reaches depends on the initial conditions. The same parameter regimes and behaviors apply for our E-I network.

The parameters used in the simulations for Fig. 4a&b are $w_0 = 0.5$, $w_1 = 2.7$, $h_0 = 10$. For the top panel, $\gamma = 0$. For 0-10s, $h_1 = 5$ and $\theta_0 = \frac{7}{4}\pi$. For 20-30s $h_1 = 5$ and $\theta_0 = \frac{3}{4}\pi$. $h_1 = 0$ otherwise, and the input is thus homogeneous.

Figure SI.1: The E and I inputs for $30\tau$, they are not balanced for network with travelling wave with the above parameters except $h_1 = 0$ throughout the simulation and $\gamma = 0.08$.

For the travelling wave, we show that the E and I inputs are not balanced, because the system does not reach equilibrium, as shown in Fig. SI.1.

## C.3 Grid attractor

The energy function is given by

$$L = -\frac{1}{2} \sum_{ij} r_i^E W_0(\mathbf{x}_i - \mathbf{x}_j) r_j^E - \sum_i A_i r_i^E + \sum_i \int_0^{r_i^E} f^{-1}(r_i^E). \tag{SI.17}$$

Here $A_i$ is the input to each neuron, in our simulation, we assume homogeneous input $A_i = A, \forall i$. $f(x) = [x]_+$ represents ReLU nonlinearity. Function $W_0(\mathbf{x})$ is given by $W_0(\mathbf{x}) = a e^{-\gamma |\mathbf{x}|^2} - e^{-\beta |\mathbf{x}|^2}$. For simulation of the spiking network with grid attractor, the parameters we used are $N_E = N_I = 63^2$, neurons are arranged in a $63 \times 63$ square sheet, $\tau_E = 0.5$, $\tau_I = 0.2$, $\tau = 0.5$, $\lambda = 6$, $\beta = 3\lambda^2$, $\alpha = 1.1$, $\gamma = 1.2\beta$, $A = 2$.

For code used to reproduce results in this paper, see `https://github.com/Pehlevan-Group/BalancedEIMinimax`.