[Reviews · NeurIPS 2020]

Review 1

Summary and Contributions: The high level context here is Excitatory Inhibitory balance in spiking neuron networks; a question of substantial interest in computational neuroscience. The earliest work in the area showed that unless you have a neuron's membrane potential hover around threshold, it is very difficult to explain its Poisson like spike train output, something that is a hallmark of cortical neurons. But what use is Excitatory Inhibitory balance? Recent previous work whose framework most fits this paper has posited that signal reconstruction (assuming a rate code) is best achieved in this regime. This paper takes a different tack. It "assumes" that the membrane potential of each neuron should hover around threshold, and then back calculates from this assumption an objective function (by setting the assumption as KKT conditions and asking what objective would have the corresponding KKT conditions). This objective is what the author(s) call the normative objective, since there is really no reason for this to be an objective, except that if this were the objective, a network whose neurons generate spikes to optimized this objective would--if it reached equilibrium--display detailed excitatory inhibitory balance. The paper then goes on to show how several other objectives that have been used by other papers can be mapped to their objective.

Strengths: The paper is fairly clearly written. The derivations are sound. Multiple applications have been shown, including what the weights of the network should be for a given input connectivity matrix for minimizing reconstruction error, as well as study of fixed point, ring and grid attractors.

Weaknesses: The biggest concern about this paper is that it is an incremental advance over: Minimax and Hamiltonian Dynamics of Excitatory-Inhibitory Networks.(1997) Seung, Rchardson, Lagarias, Hopfield, and the paper makes only passing reference to this fact Incidentally, one of the goals of the earlier paper was to incorporate Dale's law into Hopfield nets. The minimax objective is identical to the previous paper's Lyapunov function. And it suffers the same constraints as in the previous paper, W^EE and W^II are symmetric and W^IE=W^EI^T (Something that Hopfield nets have to assume to enforce fixed point dynamics, whereas there is no evidence that cortical networks satisfy this.) The claim would then be that porting the earlier paper's results to spiking neurons is by no means easy (this at the very least should have been emphasized in this paper). So it is worth considering this. The original paper has the input into a neuron as a function of the membrane potential of the presynaptic neuron, whereas this paper considers the spike train generated by the presynaptic neuron. However, at crucial places, the spike trains are replaced by their low pass filtered rate to push the analysis thru, which reduces it to the analysis in the previous model. The author rebuttal did not adequately address the the above issue. The response was that KKT conditions have additionally been derived, which to this reviewer is a small advance.

Correctness: Yes

Clarity: Yes

Relation to Prior Work: Discussion is only at a high level. Since the paper follows strongly in the lines of Minimax and Hamiltonian Dynamics of Excitatory-Inhibitory Networks.(1997) Seung, Rchardson, Lagarias, Hopfield, the author(s) should have pointed out what additional contribution they have made.

Reproducibility: Yes

Additional Feedback:


Review 2

Summary and Contributions: In this work, the authors derive balanced spiking E/I networks that obey Dale's law through the optimization of minimax objectives. Their work follows from a combination of previous studies on balanced spiking networks (Deneve & Machens 2016), as well as the construction of E/I rate networks using a similar minimax objective (Seung et al. 1998). In addition to deriving the network and optimality conditions, they show applications in signal reconstruction and various types of attractor networks.

Strengths: The paper is well written and, as far as I can tell, everything seems correct. Overall, I find that this work presents a quite intriguing way of extending the balanced spiking network formalism. First, the authors demonstrate how to design spiking networks from a loss function, in which excitatory and inhibitory neurons are separate. Extending these old (and somewhat forgotten) ideas to spiking networks is clearly a novel contribution and highly useful for the field. Second, the authors use their objective functions to design spiking networks that perform various computations. They thereby provide a new solution to an old and persistent problem in the balanced network literature, which is how to design balanced networks that perform nonlinear computations.

Weaknesses: There are no major weaknesses in the paper, but there are a few things that could be improved or corrected: (1) The minimax objectives are mostly just used here to design certain networks. However, is there a more general meaning that can be attributed to these objectives? In other words, assuming that that's how neural circuits work, why would they use minimax objectives? (2) The minimax objective allows to obtain networks that obey Dale's law, but it doesnt seem to require it (as you still have to fix the signs of the connectivity matrices). Is there any advantage to fixing those signs, or is Dale's law simply a constraint on the architecture?

Correctness: Overall all claims seem correct. One minor point: Balance is typically understood as balance of synaptic inputs, not simply the fact that voltages fluctuate around zero. In the model used here, that will also happen if a single neuron fires all by itself---but that wouldnt be called balance. So I'd suggest to use the term 'balance' more carefully.

Clarity: Yes, writing is generally very clear, given the density of the material. A few clarification questions: (1) Reconstruction of natural image patches: How many excitatory neurons were used in this example? It does not appear to be mentioned. (2) Fixed point attractors: There is not enough detail in this section. How many neurons participate in each attractor? How many attractors were stored in the network? Did the attractors include specific inhibitory neurons? Does the network oscillate? (it looks like it might based on the figure) (3) Attractor examples: It would be nice if the authors could provide some brief analysis of the weight matrices for the solutions found. E.g., do the weight matrices have any structure? Are they low-rank? How do they compare to the previous rate-based solutions to these problems?

Relation to Prior Work: Yes.

Reproducibility: Yes

Additional Feedback:


Review 3

Summary and Contributions: This study proposes a novel minimax objective function whose optimization could be implemented by a spiking neural network. In order to demonstrate the generalization and plausibility of the framework, the authors linked the new minimax objective function with the energy functions of some canonical networks, and showed their network could reproduce previous canonical networks. The main contribution of this study is the discovery of the minimax objective function.

Strengths: This study is novel in terms of that a minimax objective function was proposed, and the mathematical analysis and proof underlying the objective function are solid. The authors also showed the proposed minimax objective is able to capture some canonical network models that have been widely used in computational neuroscience.

Weaknesses: Although I believe the math derivation of the novel minimax objective function is correct, I have two major concerns. 1. My first concern is whether this minimax objective function provides some novel insight on network dynamics which cannot be captured by traditional framework that network dynamics is minimizing an “energy” function. My concern is resulted from that the minimax objective (Eq. 4) depends on the sign of each term, and which further depends on the sign of connection matrix W. Mathematically, if we absorb the minus sign in network dynamics (Eq. 2) into W (excitatory connection has positive W while inhibitory connection has negative W), we could derive an quadratic objective function similar with Eq. 4 but needs to be minimized. It seems to me that the only difference between the minimax and minimized objective function is that the network state converges to the saddle point in the former case, while in later case the network state converges to a stable fixed point. I really hope the authors explain this and correct me if I understood something wrong. Update after rebuttal: The author’s rebuttal persuaded me that the minimax objective doesn’t depend on the definition of the sign of inhibitory connection, but resulted from the anti-symmetric connections between excitatory and inhibitory neurons. 2. Another concern is about the biological plausibility of the KKT conditions in Eqs. 5-6. I certainly understand the mathematical derivations of KKT condition, but I have some difficulty in regarding this KKT condition to a real, biological neural circuit. The main obstacle is that how could we guarantee the KKT condition could be satisfied in a real neural circuit. That is, the membrane potential V is zero when firing rate is non-zero and vice versa. Moreover, the membrane potential V is smaller than the firing threshold, which is a positive number in this study, whereas Eqs. 5-6 states that V is not larger than zero. How should I understand this discrepancy? Update after rebuttal: I realize that the KKT condition is not strictly satisfied due to the discontinuous spiking network dynamics. I suggest the author explicitly state this in a revised manuscript. Moreover, to help readers understand the underlying math, the author could state which constraint in spiking dynamics leads to the KKT condition. In addition, it is not quite clear how the KKT conditions lead to the detailed balanced and tight balanced.

Correctness: I have gone through all the math in the main text, and browsed some of them in Supplementary information. I believe the math derivations I have gone through are correct and reasonable.

Clarity: The writing of this paper is structure wise, though it still could be improved if the authors could provide detailed and intuitive explanations on some key underlying assumptions. For example, 1. It is unclear what are the inequality constraints underlying the KKT conditions (Eqs. 5 and 6). Is it the spiking threshold, i.e., the membrane potential is always smaller than the spiking threshold? It would be very helpful if authors explicitly state the key underlying assumptions out. 2. I suggest the authors brief what the “variable substitution” trick means, and what is its underlying assumption. I have spent a short while to infer what this trick really means without reading the ref. [34]. Moreover, I think the underlying assumption of this trick is that only the stationary responses of r^E and r^I are considered. And the authors could add this intuitive explanation right above the Eq. 11. Some typos exist in the writing and I suggest the authors to proofread the manuscript again. 1. Eq.3: the integrand of x_j(t) should contain \tau_e instead of \tau, if I understood correctly. 2. Eq. 5, 2nd row: W^EI should be W^IE. PS: I know the authors said W^EI = W^IE (line 81), but writing a notation with this form would be easier for readers and make the notation consistent with the 2nd row in Eq. 2. 3. Eq. SI.1, 2nd row: I think the 2nd and the 3rd terms in this equation should be combined, in that they both sum over all neurons. 4. Eq. SI.1: is \delta_jk the j-th element of standard basis e_k? Please define the notation before using it. 5. Line 217: should “minimizing over r^I” be “maximizing over r^I”? 6. (Optional) Eqs. 1 and 3: Since you used notation N^0 as the number of external inputs (line 73), you may consider use s_j^0 as the external input spike to simplify notations.

Relation to Prior Work: The author discussed the connection of current study with previous work. I think the comparison could be strengthened by providing the novel, unique insight given the proposed minimax objective which cannot be obtained under traditional minimized energy function.

Reproducibility: Yes

Additional Feedback: Update after rebuttal: it seems that the minimax objective and related math derivations are all based on the firing rate. And some implicit _linear_ assumption is probably needed in deriving Eq. 2 from Eq. 1, because in general the membrane potential is not linear with firing rate, especially when the network is in the pattern forming regime. Maybe this linear relation in Eq. 2 is directly resulted from the effect of reset after every spike is absorbed in the diagonal elements in recurrent connections.

[Author Response · NeurIPS 2020]

**Reviewer 1:** We thank the reviewer for the thorough review. We agree that our discussion of Seung et al. was not sufficient and we will address this in a future version. However, our contributions go beyond Seung et al.'s work in many aspects. We kindly ask the reviewer to reconsider the following contributions. (1) Our networks are spiking, which is different from the rate dynamics of Seung et al. We derived spiking dynamics as a greedy algorithm optimizing the minimax objective. (2) We made the connection between the minimax objective, and the tight and detailed balance observed in cortical networks by investigating the KKT conditions. The notion of balanced networks is not considered at all in Seung et al., but is an important phenomenon in neuroscience (Poo et al, 2009, Rupprecht et al, 2018). (3) Normative modeling (as in Hopfield networks) has been very fruitful in this field. We extend this tool to tightly and detailed balanced spiking neural networks of excitatory and inhibitory neurons. While our framework still suffers from some of the problems of Hopfield networks, it does extend the biological plausibility of normative modeling considerably. Using our framework, we constructed balanced spiking neural networks for solving various tasks (reconstruction, fixed point and manifold attractor dynamics) heavily studied in neuroscience. The E-I circuit architectures we provide are novel to the best of our knowledge. Such applications were not available in Seung et al. We indeed applied the results in Seung et al. as a tool to provide necessary conditions of convergence of the dynamics, however, the major contribution of our paper is to demonstrate that functional balanced E-I spiking networks can be designed from minimax objectives, which is conceptually novel and potentially useful for the neuroscience community.

**Reviewer 2:** We thank the reviewer for the enthusiastic support!

*Clarification:* (1) For the image reconstruction task, the number of excitatory neurons is $N_E = 400$. (2) For the fixed point attractor, we have $N_E = 15, N_I = 15$. For the specific example we show there are only 2 attractor states stored in this network. The attractors include specific inhibitory neurons and the network weights are pretrained given these specific E and I neurons. (3) To design weights for the attractor networks, we ask the combination of the connectivities (effective weight) $\mathbf{W}^{EE} - \mathbf{W}^{EI}\mathbf{W}^{II^{-1}}\mathbf{W}^{IE}$ to mimic the weight matrices in standard attractor models, in the examples that we show, we assume uniform inhibition for simplicity, such that the structure of the effective weight is contained in $\mathbf{W}^{EE}$, the matrices $\mathbf{W}^{EE}$, $\mathbf{W}^{EI}$ and $\mathbf{W}^{IE}$ are low rank. However, there are multiple ways of designing these matrices such that the effective weight remains the same. We will provide details in the appendix.

*Minimax objectives:* We thank the author for the inspiring question. (1) Dale's law is a structural constraint for biological plausibility, it is not yet clear to us whether Dale's law has any computational advantage. (2) We proposed minimax objective as a normative approach for designing spiking networks that obey Dale's law, and we have not focused on the advantage of minimax objective itself. However, minimax objectives can be a useful strategy when faced with uncertainty, the underlying intuition behind minimax is to look for the least-worst option out of all possible outcomes of an action. Previous works have demonstrated that minimax objectives can be used to solve problems such as source localization (Venkatesh et al, 2017) and sensorimotor control (Ueyama Y.,2014), and our methods could potentially be of use to design E-I spiking neural networks that accomplish these tasks. We will discuss these points.

**Reviewer 3:** We thank the reviewer for the feedback and we will revise our paper taking all his/her suggestions into account. We kindly request a reevaluation based on the following response to the weaknesses of our paper, which we believe arose from inclarity on our part. We also ask the reviewer to consider our response to the first reviewer for what we believe are our other main contributions than the identification of the minimax objective.

*Minimax objective:* Thank you for the great question which goes to the heart of our paper! 1) Absorbing the minus sign into the definition of inhibitory weights (W) in equation (2) does NOT lead to a minimizing dynamics on a quadratic function; it still is a minimax dynamics on equation (4) with weights redefined. To see why E-I circuit dynamics cannot be gradient descent, note that gradient descent on a minimization problem would give symmetric connections (defined as the weight times the $\pm$ sign in front in our convention). However, in networks that obey Dale's law, I-to-E connections have the opposite sign of E-to-I ones. 2) We agree that the difference between gradient descent-ascent on a minimax vs. gradient descent on a minimum objective is the dynamics converging to a saddle point vs. a minimum of an objective function. However, the dynamics of approaching a saddle point can be quite different than that of a minimum, although we do not discuss these aspects in our paper. Also note that both the saddle points and the minima can be fixed points of the optimizing dynamics, which can be either discrete or form a manifold as shown in our examples. We will expand our discussion of these points.

*KKT conditions and spiking dynamics:* We thank the reviewer for giving us an opportunity to clarify this potentially confusing point. The spiking dynamics derived in SI A performs a greedy optimization of the objective function. Due to the discontinuous spiking dynamics, the KKT conditions are never exactly satisfied, consistent with the reviewer's intuition about a biological circuit. The network constantly monitors for deviation from the KKT point, and corrects for it (imperfectly) by producing spikes when $V_i$ is larger than the spiking threshold $V_{th}^i = \frac{1}{2}W_{ii}$. For the inactive (non-spiking) neurons, KKT conditions state that the voltage $V_i < 0$. This is smaller than a positive $V_{th}^i$, consistent with the neuron never spiking. We will expand our discussion of this point.

*Variable substitution:* We will clarify this. The "variable substitution trick" is merely the statement that the minimum of Eq.10 is equivalent to the saddle point of Eq.11. It holds for the optimum of the objective (or the steady state of the dynamics). We will also correct the typos the reviewer pointed out.

[Meta-Review · NeurIPS 2020]

The reviewers were originally divergent in their opinions of this paper, but came to some agreements in discussion. It was agreed that the paper provides an interesting contribution for neuroscience by extending the previous work of Seung et al. (1997) to more biologically realistic networks, but the actual theoretical insights beyond that original paper are not large. In the end, an "accept" decision was reached, but it was agreed that the authors should better clarify the strong links to the Seung paper and be more cautious in their claims of "detailed" or "tight" balance in cortical networks.